# Consistency Aware Robust Learning under Noisy Labels

**Fahad Sarfraz**                                                                                       *f.sarfraz@tue.nl*
*TomTom, Netherlands*
*Eindhoven University of Technology (TU/e), Netherlands*

**Bahram Zonooz**[*]                                                                                    *b.zonooz@tue.nl*
*Eindhoven University of Technology (TU/e), Netherlands*

**Elahe Arani**[*]                                                                                      *e.arani@tue.nl*
*Wayve Technologies Ltd, London, United Kingdom*
*Eindhoven University of Technology (TU/e), Netherlands*

**Reviewed on OpenReview:** *https://openreview.net/forum?id=pZulfLkARr*

## Abstract

Deep neural networks (DNNs) often struggle with noisy supervision, a common challenge in real-world datasets where high-quality annotations are scarce. While DNNs tend to memorize noisy labels, the human brain excels at learning in noisy environments by modulating sensitivity to errors based on their magnitude and consistency. Inspired by this, we propose Consistency-Aware Robust Learning (CARoL), which maintains a memory of past predictions and errors to quantify consistency and guide the learning process. CARoL employs a principled mechanism to distinguish clean from noisy samples and modulates rate of adaptation based on prediction consistency. Furthermore, it integrates multiple learning pathways to fully utilize the dataset, adapting to sample characteristics as training progresses. Our empirical evaluation shows that CARoL achieves high precision in noisy label detection, enhances robustness, and performs reliably under severe noise, highlighting the potential of biologically inspired approaches for robust learning. [1]

## 1 Introduction

The performance of standard deep neural networks (DNNs) is highly dependent on the availability of large amounts of high-quality annotated data. However, maintaining such levels of quality at the scale required for real-world applications is a time-intensive, resource-demanding, and costly activity. A more feasible approach is to leverage the abundant open-source web data and automate annotation using search queries and user descriptions (Makadia et al., 2008; Tsai & Hung, 2008) or to rely on crowd-sourced datasets (Mozafari et al., 2014). However, these approaches inevitably lead to label noise, which significantly degrades the performance of DNNs (Xiao et al., 2015; Zhang et al., 2016). Furthermore, DNNs are capable of memorizing even random labels (Arpit et al., 2017), adding further complexity. Given the ubiquity of noise in real-world datasets, it is critical to develop methods that can learn efficiently with noisy supervision.

As the human brain constitutes an efficient learning system that has evolved to learn under noise, we take a closer look at the dynamics of its error-based learning to draw insights for the design of DNNs. The brain modulates the rate of adaptation to error based on several characteristics, including the magnitude of the error and the uncertainty about its predictions relative to the uncertainty of its observations (Castro et al., 2014; Criscimagna-Hemminger et al., 2010; Marko et al., 2012). Essentially, the brain learns more from small, consistent errors compared to large, abrupt errors, enabled by a principled mechanism that utilizes the history of past errors(Herzfeld et al., 2014). Moreover, it is believed that small errors influence the

---

[*]Equal advisory role.
[1]Code is available at `https://github.com/NeurAI-Lab/CARoL`

process of learning in a fundamentally different manner than large errors (Criscimagna-Hemminger et al., 2010). We believe that these salient features and learning mechanisms enable the brain to learn effectively under high levels of supervision noise and uncertainty in the environment. Hence, distilling these properties into DNNs can improve their robustness to label noise.

To this end, we first analyze how learning progresses under high degrees of label noise with standard training. Specifically, we examine how predictions change for samples with clean versus noisy labels and define consistency as the fraction of predictions across training epochs that agree with the provided training label. Our analysis reveals that clean samples maintain high prediction consistency and alignment with the training label throughout training, while noisy samples initially exhibit low consistency values, which gradually increase as the model begins memorizing the noisy labels. Interestingly, this memorization occurs earlier for noisy labels that are semantically similar to the true label. Furthermore, early predictions for noisy samples often align with their true labels. This observation suggests that if memorization can be avoided, the model's predictions during the early phase of training can serve as a good surrogate for true labels, providing a strong incentive to bias learning towards this phase. These findings provide a systematic mechanism to quantify the phenomenon that DNNs learn simple patterns before memorizing noisy labels (Arpit et al., 2017). These insights underscore the importance of using instance-level statistics rather than population-level thresholds to better differentiate between clean and noisy labels, guide the development of more effective noise-detection criteria, estimate noise transition matrices, and design robust training mechanisms.

Drawing insights from our empirical analysis and inspired by the learning dynamics of the brain, we propose *Consistency-Aware Robust Learning (CARoL)*, a framework designed to enhance the robustness of deep neural networks under noisy supervision. CARoL maintains a memory of past predictions and errors during training, leveraging both consistency between predictions and labels, along with error magnitude, to guide the learning process. It quantifies prediction consistency across training epochs for each sample and uses this information, along with error magnitude, to dynamically adjust the learning process, distinguishing between clean and noisy samples. Based on this, we introduce a novel consistency-aware error modulation loss, which adjusts the contribution of each sample to the training objective by weighting samples

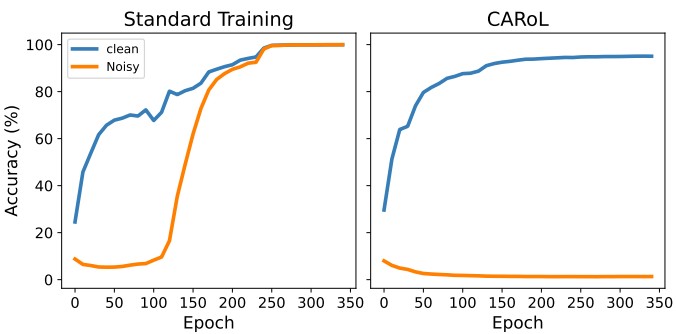

Figure 1: Training accuracy on clean and noisy samples on CIFAR-10 with 80% symmetric noise. While standard training memorizes the noisy labels, even at such severe noise levels, our method, (*CARoL*), effectively avoids memorization.

according to their consistency and their deviation from expected loss patterns. This ensures that reliable samples contribute more significantly to the model's optimization while noisy samples are downweighted. To further enhance robustness, CARoL integrates multiple learning pathways, leveraging a combination of supervised and auxiliary learning strategies to handle diverse samples effectively. For reliable samples, CARoL prioritizes supervised learning to focus on extracting clean patterns. For uncertain or noisy samples, it employs auxiliary mechanisms that guide the model to learn useful representations without relying heavily on noisy labels. Additionally, CARoL incorporates strategies to encourage diversity in training and maintain consistency in predictions over time, ensuring that learning remains robust across different phases of training. These pathways are dynamically adapted based on the training phase, enabling the model to focus on generalizable patterns during early training and progressively refine its robustness to noise.

We empirically demonstrate the effectiveness of our approach across a range of challenging noisy supervision scenarios. Our proposed selection criterion achieves high precision in detecting noisy labels and ensures a balanced selection of samples across classes. Combined with the dynamic rate of adaptation to errors and the integration of multiple learning pathways, CARoL exhibits exceptional robustness to supervision noise

and avoids memorization, even under severe label noise conditions (Figure 1). We believe our work paves the way for further exploration of biologically inspired mechanisms in robust learning under noisy supervision.

**Our key contributions are as follows:**

- We introduce **Consistency-Aware Robust Learning (CARoL)**, which leverages a memory of past predictions to define per-sample consistency and uses this as the guiding criterion for learning under noisy labels.

- We propose a **class-adaptive thresholding mechanism** that adjusts consistency thresholds per class relative to the global average, accounting for varying class difficulty and noise prevalence.

- We design a biologically inspired **Consistency-Aware Error Adaptation loss** ($\mathcal{L}_{CA}$), which modulates sample weights based on error magnitude and consistency to suppress memorization while preserving informative signals.

- We integrate **multiple learning pathways**—supervised learning for consistent samples, SSL via FixMatch for inconsistent samples, and a novel consistency-aware MixUp—to effectively leverage all training data.

- Through comprehensive experiments on CIFAR-10/100, Tiny-ImageNet, and real-world noisy datasets (Web-Aircraft, Web-Bird, Web-Car), we demonstrate that CARoL effectively mitigates memorization and effectively learns under severe noise and in complex real-world scenarios.

## 2 Related Work

The different approaches to learning under noisy labels can be broadly categorized into three categories: Estimate the noise transition matrix to relabel training data (Tanaka et al., 2018; Patrini et al., 2017; Hendrycks et al., 2018), improve the robustness of the loss function to label noise (Zhang & Sabuncu, 2018; Ma et al., 2020), and separate noisy and clean samples to inform learning (Arazo et al., 2019; Ding et al., 2018; Han et al., 2018). Among these, methods that attempt to identify noisy samples and employ a semi-supervised (Zhu, 2005) learning approach have proven to be effective, particularly for high degrees of label noise and a larger number of classes.

The vast majority of these methods rely on the low-loss criterion, whereby samples with low training loss are considered clean. To reduce confirmation bias, CoTeaching (Han et al., 2018) trains two models alternately, each model selecting a predefined percentage of samples with the smallest loss as clean samples for the other model. M-Correction (Arazo et al., 2019) and DMix (Li et al., 2020) models the loss distribution with a mixture model to split the training data into clean and noisy samples and train the two models in a semi-supervised manner. MOIT (Ortego et al., 2021) combines semi-supervised learning with contrastive learning. JPL (Kim et al., 2021) proposes negative learning for noisy labels. ELR (Liu et al., 2020) regularizes the update of the model that utilizes the early phase of learning. UNICON (Karim et al., 2022) highlights the issue of class imbalance in the low loss selection criterion and proposes a Jansen-Shannon divergence-based uniform selection mechanism. Importantly, the small-loss trick usually requires an accurate estimate of noise level or biases learning toward easy samples, as harder samples are likely categorized as noisy. Instead of separating samples, NCT (Sarfraz et al., 2021), improves the robustness of the learning framework by leveraging collaborative learning and target variability. SOP (Liu et al., 2022) models label noise as a sparse component and uses implicit regularization to separate it from clean data, improving robustness in over-parameterized networks. Co-LDL (Sun et al., 2021a) incorporates low loss sample selection strategy with label distribution learning. RML-Semi (Li et al., 2024) mitigates label noise by combining stable mean and robust median losses via a regrouping strategy, improving sample selection and enabling a semi-supervised learning approach.

Closest to our approach, SELFIE (Song et al., 2019), also employs prediction consistency. However, they solely use the model's past predictions within a time window to correct the noisy labels and continue to employ the small-loss trick to differentiate between clean and noisy samples, assuming the availability of the noise rate, which is not known a priori (detailed comparison provided in Appendix). Similarly RoCL (Zhou

et al., 2020) and CARoL share the intuition that accumulated signals are more reliable than instantaneous ones, their approaches differ fundamentally. RoCL relies primarily on small-loss detection (applied to EMA of per-sample loss) and uses consistency only for pseudo-label generation (detailed comparison in Appendix). In comparison to these approaches, our study introduces prediction consistency as a more sturdy and reliable indicator of label quality and dynamically adapts the contribution of each sample based on their consistency value and error magnitude. We draw inspiration from the error-based learning dynamics of the brain to guide our learning process.

## 3   Evolution of Predictions during Training

To gain a deeper understanding of the learning dynamics and prediction patterns of DNNs under high levels of label noise, we examine how model predictions evolve during training and differ for samples with clean and noisy labels. Specifically, we track the predictions of each sample at each epoch during training and measure consistency by evaluating the fraction of predictions that align with the provided training label. Figure 2 plots the consistency in predictions at different stages of learning for the PreAct-ResNet18 model He et al. (2016) trained with the standard cross-entropy loss on CIFAR-10 with 50% symmetric label noise. During the early stages of training, samples with clean labels (clean samples) exhibit significantly higher prediction consistency (skewed toward 1), whereas samples with noisy labels (noisy samples) show very low consistency (skewed toward 0), indicating poor alignment with the training labels. Although clean samples maintain high consistency throughout training, noisy samples progressively memorize the incorrect label during the middle phase, ultimately reaching a median consistency value of 0.5 by the end of training. These observations help characterize how predictions evolve during training and offer a systematic way to quantify the phenomenon where DNNs learn simple patterns before memorizing noisy labels (Arpit et al., 2017).

For noisy samples, the model predictions are predominantly split between the true and incorrect training labels. Figure 3 (a) illustrates the degree of alignment between the most predicted class and both the training and true labels. Interestingly, during the initial epochs, the model often predicts the true label with high consistency, even for noisy samples. As training progresses, however, the model begins memorizing noisy labels, and the most predicted class becomes increasingly aligned with them. This highlights the shift in prediction dynamics as the model transitions from learning general patterns to memorizing noisy data.

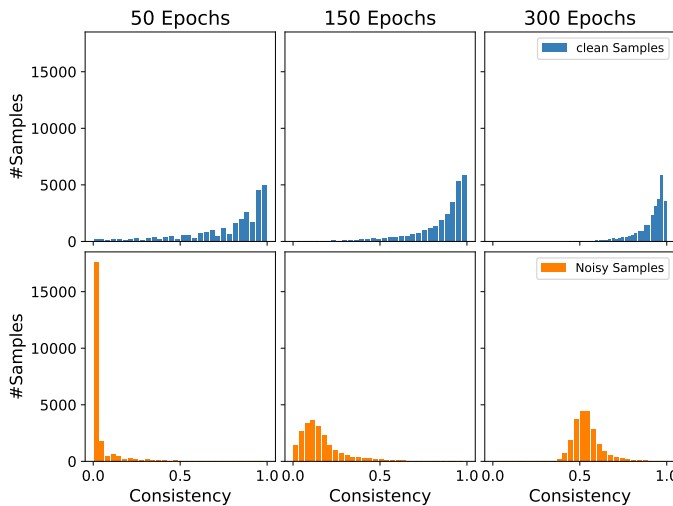

Figure 2: Distribution of the percentage of predictions that align with the training label for clean samples and noisy samples.

We also observe that the onset of memorization for noisy samples varies for each instance depending on the semantic similarity between the true label and the noisy label. Figure 3 (b) shows that in cases where the noisy label is semantically similar to the true label (e.g., cat and dog, automobile and truck), memorization occurs earlier, as reflected by a higher average consistency value. Early memorization results in a larger fraction of predictions aligning with the noisy label. In contrast, when the noisy label is dissimilar to the true label (e.g. auto and cat), memorization occurs much later. These findings suggest that instance-level statistics aggregated over training and adaptive classwise thresholds can be leveraged to design distinguishing criteria, noise transition matrix estimation, and robust training mechanisms.

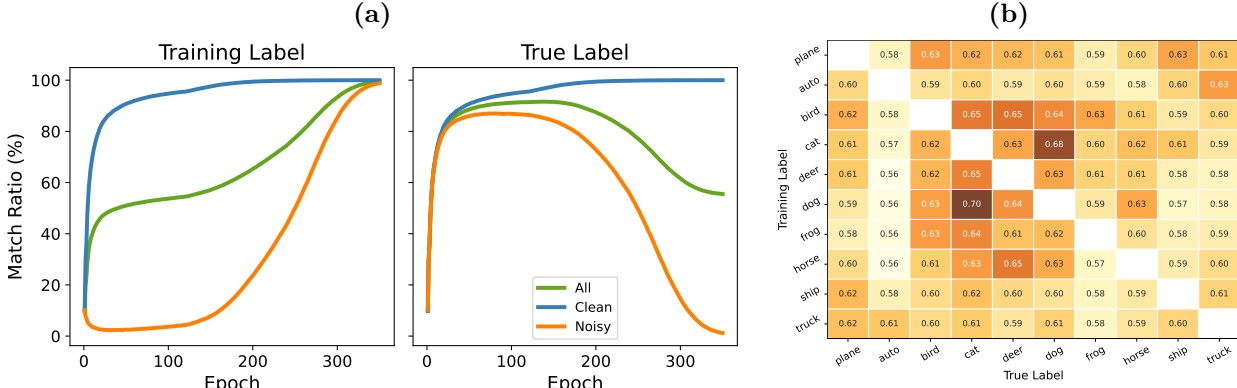

Figure 3: (a) Percentage of samples in which the most frequently predicted class aligns with the training label (left) or the true label (right) during the training process. (b)Average consistency for each noisy training label and the true label pair at the end of training. Diagonal values (clean samples) are omitted for better visualization.

## 4 Methodology

We motivate our method using insights from human brain learning dynamics and our analysis of DNN prediction patterns, before detailing its components.

### 4.1 Overview

The design of CARoL is motivated by the observation that the brain prioritizes learning from small, consistent errors over large, abrupt ones. Inspired by this principle, CARoL maintains explicit memories of both a model's past predictions and its classwise error statistics. Consistency is defined as the percentage of predictions that match the training label across the trajectory, while the error memory captures class-dependent loss dynamics. Together, these memories provide the basis for robust and informed sample weighting.

Our analysis reveals that clean samples maintain high consistency across training, while noisy samples exhibit low consistency early on but gradually increase as the model memorizes the noisy label. Importantly, early predictions for noisy samples often align with their true labels, and the speed of memorization depends on semantic similarity (e.g., "dog" mislabeled as "cat" memorizes earlier than "dog" mislabeled as "car"). These insights suggest that early-phase predictions are a valuable surrogate for ground truth if memorization is avoided.

CARoL operationalizes these findings through three mechanisms: (i) an EMA model biased toward early-phase predictions to preserve reliable signals, (ii) class-adaptive thresholds to account for class-dependent noise patterns, and (iii) a principled error sensitivity modulation scheme that uses prediction and error memories to down-weight inconsistent samples based on their deviation from classwise error statistics. Furthermore, CARoL leverages all training data by combining supervised learning for consistent samples, SSL with EMA pseudo-labels for inconsistent samples, and a consistency-aware MixUp objective. This unified and principled design enables CARoL to effectively mitigate memorization while extracting useful information from both clean and noisy samples.

### 4.2 Biological Basis of Consistency as a Learning Cue

Consistency plays a vital role in error-based learning in the brain, enabling it to reliably predict future events and understand cause-and-effect relationships to build an internal model of the world. The brain adjusts its sensitivity to errors based on factors such as error magnitude and the gap between prediction

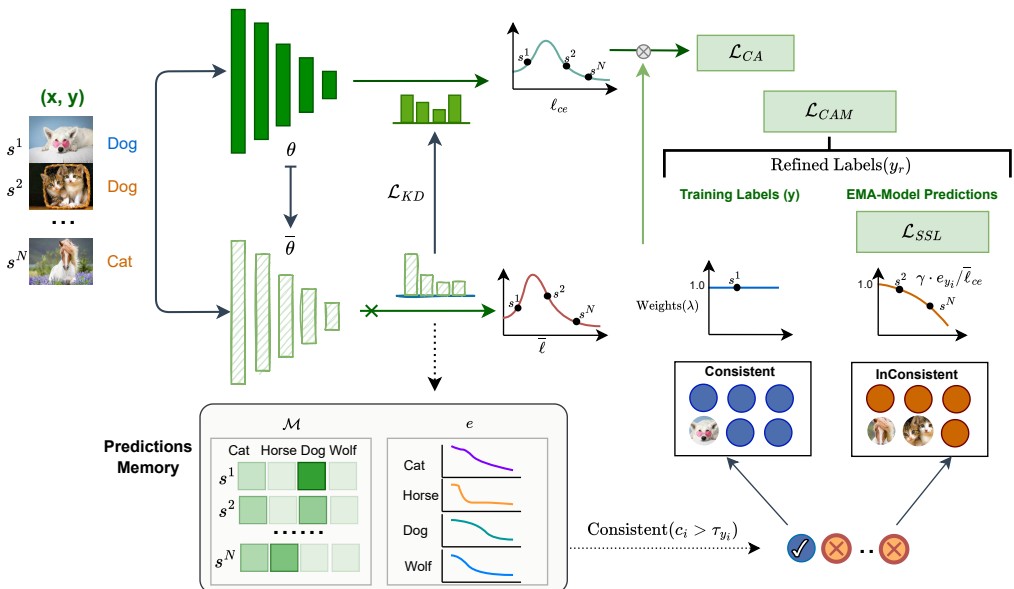

Figure 4: CARoL: A consistency-aware robust learning method that incorporates a principled mechanism to learn from consistent small errors and adapt learning based on sample characteristics. It uses an exponentially moving averaged (EMA) model to accumulate knowledge from different stages of training and maintains a memory of past predictions. CARoL utilizes consistency score to form consistent set (blue colored samples) and inconsistent set (orange samples) as a proxy for clean and noisy samples. Subsequently, the weight of inconsistent samples is reduced proportionally to their distance from the classwise error statistics. In addition, it uses a combination of semi-supervised learning and consistency-aware MixUp loss to maximize learning from all samples.

and observation uncertainty (Castro et al., 2014). These factors are closely tied to consistency: small errors suggest that feedback aligns well with the brain's internal model, while lower prediction uncertainty reflects greater alignment of the stimulus with its prototypical representations. Consequently, the brain prioritizes learning from small, consistent errors over large, abrupt ones, adapting its learning process to focus on reliable patterns and filter out noisy or inconsistent information (Criscimagna-Hemminger et al., 2010; Marko et al., 2012).

Additionally, evidence suggests that the brain moderates its learning from present errors via a systematic mechanism that accounts for the history of previous errors, requiring the retention of an error memory (Herzfeld et al., 2014). Small errors also appear to influence learning in fundamentally distinct ways compared to large errors, as studies show that the neural basis of motor learning differs between these two cases (Criscimagna-Hemminger et al., 2010). We hypothesize that these mechanisms enable the brain to learn effectively in noisy environments, and we aim to integrate similar strategies into DNNs.

## 4.3 Consistency-Aware Robust Learning

Building on our analysis of prediction evolution and the brain's reliance on consistency for learning, we hypothesize that instance-level consistency in predictions can serve as a crucial cue for DNNs. It enables robust identification of label noise while also providing insights into a sample's learning difficulty, its coherence with other samples in the same class, and the overall reliability of its label. Inspired by the brain's ability to regulate learning through error sensitivity, CARoL maintains a memory of past predictions and errors to assess consistency as training progresses. To further refine adaptation, it accounts for variations in class difficulty through dynamic classwise thresholds, ensuring systematic calibration at both the instance and class levels. By leveraging both prediction history and classwise adjustments, CARoL modulates adaptation

rates based on prediction consistency and relative error magnitude, allowing the model to prioritize reliable patterns while mitigating memorization of noisy labels.

### 4.3.1 Components

CARoL trains a working model, $f_\theta$, on a batch of training samples $\boldsymbol{X}$ with labels $\boldsymbol{y}$, which may include noisy labels. To capture evolving knowledge and mitigate confirmation bias, we maintain an exponentially moving averaged (EMA) model (Tarvainen & Valpola, 2017), $f_{\bar{\theta}}$, updated using $f_\theta$'s weights. The EMA model evolves smoothly, preserving early-stage knowledge, and provides a stable representation of the data distribution. We use a stochastic update strategy (B.1) to prioritize early-stage information, reducing the model's susceptibility to memorization. This complementary perspective aids in regulating adaptation and decision-making throughout training.

### 4.3.2 Consistency in Predictions

The variance in model predictions during training provides valuable insights into the quality of training labels and sample difficulty. To capture this, we maintain a per-sample prediction memory that tracks how often the EMA model assigns each label. Let $\boldsymbol{M} \in \mathbb{R}^{N \times K}$ denote this memory, where $N$ is the number of samples and $K$ is the number of classes. The prediction consistency vector, $\boldsymbol{c} \in \mathbb{R}^N$, quantifies the proportion of EMA model predictions that match the training label $\boldsymbol{y}$. For a sample with index $i$, the consistency score is given by:

$$c_i = \frac{M_{i,y_i}}{\sum_{j=1}^{K} M_{i,j}} \tag{1}$$

We use the consistency score to distinguish between noisy and clean samples. To account for variations in class difficulty, we adapt the threshold hyperparameter $\tau$ based on the class-wise average consistency, computed as the mean consistency of all samples in class $k$, and the global average consistency across all samples.

$$\tau_k = \frac{N}{n_k} \cdot \frac{\sum_{i=1}^{N} \mathbb{I}(y_i = k)c_i}{\sum_{i=1}^{N} c_i} \cdot \tau, \tag{2}$$

where $n_k$ is the number of samples in class $k$. A sample with index $i$ is assigned to the consistent set $\mathbb{C}$ if its consistency value is higher than the class-wise threshold:

$$\mathbb{C} = \{i \mid c_i \geq \tau_{y_i}, \ \forall i \in N\}. \tag{3}$$

Furthermore, we integrate prediction consistency and error magnitude to ensure a more balanced selection of samples in the consistent set across classes, dynamically adjusting selection to mitigate class imbalance (see B.2).

### 4.3.3 Consistency-Aware Error Adaptation

To emulate the brain's ability to adjust its sensitivity to errors, we track class-wise prediction error statistics over training and leverage this information to dynamically weight each sample in the supervised learning objective. Concretely, we maintain a class-wise error memory, $\boldsymbol{e} \in \mathbb{R}^K$, which is updated using an exponential moving average of the EMA model's mean cross-entropy loss per class in each training batch.

We then apply a consistency-aware weighting scheme, assigning a high weight to samples in the consistent set $\mathbb{C}$ while scaling down others based on their deviation from mean class-wise loss statistics. For each sample with index i, the weight is given by:

$$\lambda_i = \begin{cases} 1 & \text{if } i \in \mathbb{C} \\ \min\left(1, \ \frac{\gamma \cdot \boldsymbol{e}_{y_i}}{\ell_{ce}(f(\boldsymbol{X}_i; \bar{\theta}), \boldsymbol{y}_i)}\right) & \text{otherwise} \end{cases} \tag{4}$$

---

**Algorithm 1** Consistency-Aware Robust Learning (CARoL) Algorithm

---

**Input:** Training dataset ($\mathcal{D}$), learning rate ($\eta$) and hyperparameters: ($\tau$, $\gamma$)
**Initialize:** $\bar{\theta} = \theta$, $\boldsymbol{M} = \text{Zeros}(\mathbb{R}^{N \times K})$, $\boldsymbol{e} = \text{Zeros}(\mathbb{R}^K)$
**Warm Up Stage:** Train on CE loss and update prediction and error memory.
**while** Training **do**
    **Identify Consistent Samples:** based on consistency in past predictions (Eq. 3).
    **Consistency Aware Learning:**
    Sample batch: $(\boldsymbol{X}, \boldsymbol{X}'', \boldsymbol{y}) \sim \mathcal{D}$
    Compute: $\boldsymbol{z} = f(\boldsymbol{X}; \theta)$, $\bar{\boldsymbol{z}} = f(\boldsymbol{X}; \bar{\theta})$
    Evaluate weights $\lambda$ using consistency, memory, and $\ell_{ce}(\bar{\boldsymbol{z}}, \boldsymbol{y})$   (Eq. 4)
    Combine losses: $\mathcal{L} = (1 - \beta)\mathcal{L}_{CA} + \beta\mathcal{L}_{KD} + \mathcal{L}_{SSL} + \mathcal{L}_{CAM}$
    Update model: $\theta \leftarrow \theta - \eta\nabla_\theta\mathcal{L}$
    Update EMA: $\bar{\theta} \leftarrow \alpha\bar{\theta} + (1 - \alpha)\theta$ if $r_F > u \sim U(0, 1)$   (Eq. 7)
    Update memories: $\text{Update}(\boldsymbol{M}, \arg\max(\bar{\boldsymbol{z}}))$, $\text{Update}(\boldsymbol{e}, \ell_{ce}(\bar{\boldsymbol{z}}, \boldsymbol{y}))$  (Eq. 9)
**end while**
**Return:** $\theta$

---

where $\gamma$ is a scaling factor and $\ell_{ce}$ is the cross-entropy loss. Subsequently, our proposed consistency-aware loss is evaluated as the weighted sum of all sample losses:

$$\mathcal{L}_{CA} = \sum_i \lambda_i \ \ell_{ce}(f(\boldsymbol{X}_i; \theta), \boldsymbol{y}_i) \tag{5}$$

By integrating prediction consistency and error adaptation, this approach enables us to incorporate cues about the quality of supervision labels and bias the learning process towards small consistent errors, enhancing the model's robustness to noisy labels and reduce memorization.

### 4.3.4 Semi-Supervised Learning

Inspired by the brain's ability to process small, consistent errors differently from large, uncertain ones, we employ distinct learning mechanisms based on a sample's consistency relative to past predictions. While samples with small, consistent errors are primarily learned using supervised loss, inconsistent samples (most likely noisy) are leveraged through semi-supervised learning. We adapt the FixMatch loss(Sohn et al., 2020) by using the EMA model's predictions on weakly augmented images as pseudo-labels to train the model with semi-supervised loss, $\mathcal{L}_{SSL}$, on the strongly augmented version of the same image. Additionally, to promote consistency across training and incorporate knowledge from the early learning phase, we employ structural knowledge distillation (Hinton et al., 2015). Specifically, we use KL-divergence loss, $\mathcal{L}_{KD}$, between the predictions of the working model and the EMA model. As the EMA model aggregates more knowledge from earlier phases of training, this biases learning toward stable patterns captured before the model starts memorizing noisy labels, acting as both a form of guidance and regularization.

By integrating these mechanisms, we exploit the early phase of training, where the model is less susceptible to memorization, while simultaneously utilizing inconsistent samples to enhance representation learning.

### 4.3.5 Consistency-Aware MixUp

To leverage all samples while mitigating the impact of noisy labels, we employ a consistency-aware MixUp strategy. For samples in the consistent set $\mathbb{C}$, we use their given labels, while for the remaining samples, we assign pseudo-labels $\boldsymbol{p}$ derived from the EMA model's predictions. Since the EMA model aggregates knowledge over training and is less prone to memorization, it often provides more reliable labels for noisy samples.

We then apply MixUp augmentation, where each sample is interpolated with a randomly selected sample from the batch, blending both their inputs and refined labels using a mixing ratio sampled from a Beta distribution. This increases training diversity and enables a supervised learning signal that is more robust to

Table 1: Comparison with prior methods on CIFAR-10 and CIFAR-100 datasets with **symmetric noise**. Baseline results are obtained from the respective articles.

| | CIFAR-10 | | | CIFAR-100 | | |
|---|---|---|---|---|---|---|
| Method | 20% | 50% | 80% | 20% | 50% | 80% |
| CE | 86.8 | 79.4 | 62.9 | 62.0 | 46.7 | 19.9 |
| M-Up | 95.6 | 87.1 | 71.6 | 76.8 | 57.2 | 30.8 |
| PCI | 92.4 | 89.1 | 77.5 | 69.4 | 57.5 | 31.1 |
| JPL | 93.5 | 90.2 | 35.7 | 70.9 | 67.7 | 17.8 |
| MOIT | 94.1 | 91.1 | 75.8 | 75.9 | 70.1 | 51.4 |
| DMix | 96.1 | 94.6 | 92.9 | 77.3 | 74.6 | 60.2 |
| ELR+ | 95.8 | 94.8 | 93.3 | 77.6 | 73.6 | 60.8 |
| UniCon | 96.0 | 95.6 | **93.9** | 78.9 | 77.6 | 63.9 |
| SOP+ | 96.3 | 95.5 | **94.0** | 78.8 | 75.9 | 63.3 |
| RML-Semi | **96.5** | 95.7 | 93.9 | 78.9 | 77.8 | 64.1 |
| CARoL | 96.3 | **95.9** | 93.8 | **79.7** | **77.9** | **66.0** |

Table 2: Comparison with prior methods on Tiny-ImageNet dataset with symmetric and asymmetric (pair flip) label noise. Baseline results are obtained from respective articles and (Karim et al., 2022).

| Noise Type | Symmetric | | | | Asymmetric | |
|---|---|---|---|---|---|---|
| Noise (%) | 20 | | 50 | | 45 | |
| Method | Best | Avg. | Best | Avg. | Best | Avg. |
| CE | 35.8 | 35.6 | 19.8 | 19.6 | 26.3 | 26.2 |
| Decoupling | 37.0 | 36.3 | 22.8 | 22.6 | 26.6 | 26.1 |
| MentorNet | 45.7 | 45.5 | 35.8 | 35.5 | 26.6 | 26.2 |
| Co-teaching+ | 48.2 | 47.7 | 41.8 | 41.2 | 26.9 | 26.5 |
| M-correction | 57.2 | 56.6 | 51.6 | 51.3 | 24.8 | 24.1 |
| NCT | 58.0 | 57.2 | 47.8 | 47.4 | 43.0 | 42.4 |
| UNICON | 59.2 | 58.4 | 52.7 | 52.4 | - | - |
| CARoL | **61.0** | **60.8** | **55.2** | **54.9** | **47.3** | **45.2** |

label noise while reducing the risk of memorization. The resulting samples are trained using the consistency-aware MixUp loss, $\mathcal{L}_{CAM}$, which further improves model robustness under noisy supervision.

### 4.3.6 Overall Loss

The overall training objective is given by:

$$\mathcal{L} = (1 - \beta)\mathcal{L}_{CA} + \beta\mathcal{L}_{KD} + \mathcal{L}_{SSL} + \mathcal{L}_{CAM} \tag{6}$$

where $\beta$ controls the relative contribution of the consistency-aware loss and the distillation loss. Following prior work (Sarfraz et al., 2021), we progressively increase $\beta$ from 0 to 0.95 during training using a sigmoid ramp-up function, gradually shifting the focus towards distillation. The overall formulation of CARoL is outlined in Algorithm 1.

Additional implementation details are provided in the Appendix. For inference, we use only the working model $f_\theta$, foregoing the commonly used ensemble approach. This ensures computational efficiency while maintaining strong performance, making our method well-suited for real-world deployment.

## 5 Experimental Setup

We evaluate our method across diverse simulated and real-world noisy datasets with varying noise levels. Following prior works (Karim et al., 2022; Li et al., 2020), we introduce symmetric noise on CIFAR-10,

Table 3: Comparison with SOTA approaches in test accuracy (%) on real-world noisy datasets: Web-Aircraft, Web-Bird, Web-Car. Baseline results are from (Sun et al., 2021a)

| Methods | Web-Aircraft | Web-Bird | Web-Car | Average |
|---|---|---|---|---|
| CE | 60.80 | 64.40 | 60.60 | 61.93 |
| Decoupling | 75.91 | 71.61 | 79.41 | 75.64 |
| Co-teaching+ | 74.80 | 70.12 | 76.77 | 73.90 |
| PCIL | 78.82 | 75.09 | 81.68 | 78.53 |
| DMix | 82.48 | 74.40 | 84.27 | 80.38 |
| Self-adaptive | 77.92 | 78.49 | 78.19 | 78.20 |
| Co-LDL | 81.97 | **80.11** | 86.95 | 83.01 |
| CARoL | **88.21** | 78.10 | **88.58** | **84.96** |

Table 4: Effect of the different components of CARoL on performance on CIFAR-10 with multiple levels of symmetric label noise.

| $\mathcal{L}_{KD}$ | $\mathcal{L}_{CA}$ | $\mathcal{L}_{SSL}$ | $\mathcal{L}_{CAM}$ | 50% | | 80% | |
|---|---|---|---|---|---|---|---|
| | | | | BEST | LAST | BEST | LAST |
| ✗ | ✗ | ✗ | ✗ | 82.5 | 59.1 | 66.0 | 28.1 |
| ✓ | ✗ | ✗ | ✗ | 87.8 | 60.4 | 71.2 | 28.3 |
| ✗ | ✓ | ✗ | ✗ | 88.6 | 77.2 | 73.9 | 44.8 |
| ✓ | ✓ | ✗ | ✗ | 90.7 | 88.2 | 77.4 | 52.9 |
| ✓ | ✓ | ✓ | ✗ | 92.1 | 92.5 | 84.5 | 84.1 |
| ✓ | ✓ | ✗ | ✓ | 95.3 | 95.1 | 92.8 | 92.6 |
| ✓ | ✓ | ✓ | ✓ | **95.9** | **95.5** | **93.8** | **93.5** |

CIFAR-100, and TinyImageNet by randomly replacing labels with uniformly sampled incorrect labels. Additionally, we model asymmetric noise on TinyImageNet (Sarfraz et al., 2021; Yu et al., 2019) using the pair-flip strategy, which simulates annotation errors where visually similar classes are confused. To assess real-world applicability, we evaluate on three fine-grained datasets: Web-Aircraft, Web-Bird, and Web-Car (Sun et al., 2021b). These datasets contain 13,503, 18,388, and 21,448 noisy training images across 100, 200, and 196 categories, respectively, with clean test sets for evaluation. The training images are curated from web search engines, which contains arbitrary levels of noise.

Training details are provided in Appendix C. The selected hyperparameters are listed in Table 5, while Table 6 shows that performance remains stable across various parameter choices. Notably, CARoL's hyperparameters exhibit a synergistic relationship, allowing most parameters to remain fixed, thereby simplifying hyperparameter optimization.

# 6 Empirical Evaluation

We evaluate CARoL across multiple benchmarks under uniform experimental settings, comparing it with state-of-the-art (SOTA) methods under varying noise levels and types. Table 1 demonstrates that CARoL achieves either superior or competitive performance across different levels of symmetric label noise on CIFAR-10 and CIFAR-100. Notably, CARoL shows substantial gains in the most challenging setting—CIFAR-100 with 80% noise which represents severe levels of label noise on a complex dataset with a higher number of semantically similar classes. Table 2 further highlights CARoL's effectiveness on TinyImageNet, a more complex dataset with 200 classes, where it significantly outperforms prior methods under both symmetric and asymmetric noise conditions.

Crucially, the performance gains become more pronounced as dataset complexity increases, demonstrating CARoL's robustness to label noise in scenarios with high inter-class similarities. These improvements are particularly notable given that CARoL achieves them with a single model and single-model inference, unlike many strong baselines that rely on concurrent training of multiple models and ensemble inference. For

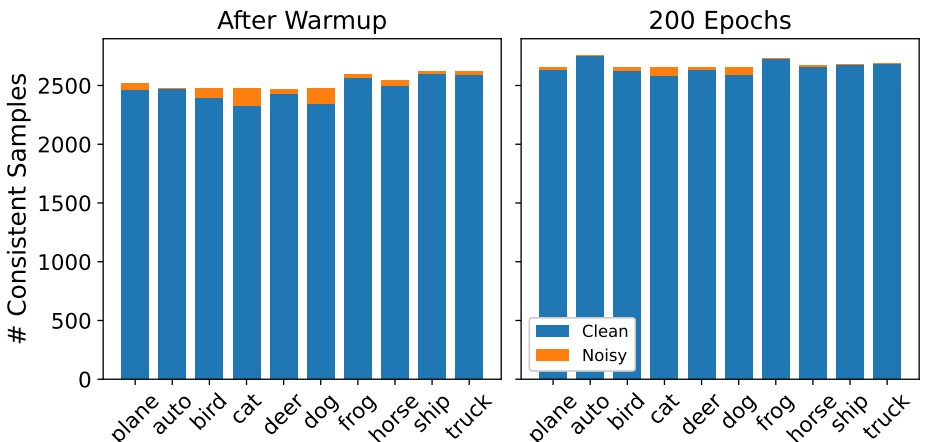

Figure 5: The distribution of clean and noisy labels among the samples from each class that are considered consistent, measured after the warm-up stage (30 epochs) and after 200 epochs. Furthermore, Figure 7 in Appendix shows that CARoL effectively identifies the vast majority of noisy samples as inconsistent, reducing false negatives.

instance, the one-model variant of ELR (ELR in Table 1 (Liu et al., 2020)) only achieves 80.7% on CIFAR-10 (80% noise) and 30.3% on CIFAR-100 (80% noise), significantly lower than CARoL's performance.

To further assess CARoL's applicability, we evaluate it on real-world noisy datasets (Table 3). CARoL demonstrates strong performance compared to prior works, providing significant gains in Web-Aircraft and Web-Car while remaining competitive on Web-Bird. While real-world datasets introduce additional challenges due to the non-uniform nature of label noise, CARoL's consistency-aware learning mechanism enables it to remain robust across different noise structures.

Overall, CARoL's consistency-aware adaptation allows it to effectively utilize all training samples without rigidly separating clean and noisy instances, making it highly robust across various noise conditions. Unlike methods that rely on explicit noise assumptions or strong clean/noisy sample separation, CARoL dynamically modulates learning signals, ensuring scalability and real-world applicability.

# 7 Ablation Study

To gain insights into the learning mechanisms in CARoL, we conduct a stepwise ablation, incrementally adding each objective function to the standard cross-entropy (CE) loss and evaluating its impact on model performance. Table 4 shows that each component contributes positively, with their effect becoming more pronounced at higher noise levels. In particular, the Consistency-Aware Error Adaptation loss ($\mathcal{L}_{CA}$) significantly mitigates memorization, underscoring its ability to identify noisy samples and suppress memorization by down-weighting inconsistent samples. The benefit is further amplified when combined with $\mathcal{L}_{KD}$, showing the complementary role of consistency-based modulation and EMA-guided regularization.

The Consistency-Aware Error Adaptation loss significantly improves the model's ability to extract meaningful supervision from noisy labels while mitigating memorization. This is complemented by semi-supervised learning (SSL) and Consistency-Aware MixUp, both of which enable the model to leverage all training samples effectively. The SSL loss utilizes the EMA model to learn from confident predictions on the inconsistent samples, improving overall performance. Notably, Consistency-Aware MixUp refines target labels by leveraging the EMA model's predictions, facilitating a smoother interpolation between samples and further enhancing generalization. Overall, our ablation study demonstrates the benefits of dynamically adapting learning objectives based on prediction consistency and error magnitude.

# 8 Quality of Selection Criterion

One of the key challenges in learning under noisy labels is accurately distinguishing between clean and noisy samples to guide the learning process effectively. Therefore, we evaluate the effectiveness of our selection criterion in identifying clean samples. Figure 5 plots the number of clean and noisy samples per class that are classified as consistent ($i \in \mathbb{C}$) using our approach when training on CIFAR-10 with 50% noise. Even in the early training stages (after the warm-up phase), our method correctly identifies the majority of clean samples while minimizing false positives.

Furthermore, our class-wise consistency threshold accounts for varying class difficulty, leading to a more balanced selection across classes. Unlike the small-loss criterion Karim et al. (2022), which can exacerbate class imbalance by favoring easy-to-learn classes, our selection criterion ensures a more uniform distribution of selected samples across all classes. These results support our hypothesis that tracking prediction consistency over training provides a reliable signal for assessing supervision quality, offering a principled alternative to loss-based sample selection.

# 9 Conclusion

Our study analyzed the evolution of the predictions during training and compared the consistency of the predictions for clean and noisy samples to provide new insights into the behavior of the model under label noise. We then utilized these findings and insights from the brain to design a novel method, Consistency-Aware Robust Learning, which provides a principled approach for utilizing the evolution of instance-level consistency in predictions of samples and error magnitude to estimate the quality of supervision and accordingly employs different learning mechanisms for efficient learning under high degrees of label noise. The empirical evaluation demonstrated the effectiveness of our approach on various datasets with different types and degrees of label noise and showed that it improves precision in detecting noisy labels and enables the model to learn effectively even under severe levels of label noise.

# 10 Broader Impact Statement

Learning under label noise is critical for real-world applications, where large-scale datasets are often constructed through automated labeling, inevitably introducing annotation errors. Effectively handling such noise enables the utilization of vast, imperfect data sources, reducing dependence on costly manual annotations and improving the scalability of DNNs. CARoL demonstrates the potential of incorporating biologically inspired mechanisms for robust learning by incorporating principles of error-based learning, where the brain dynamically modulates sensitivity to errors based on their magnitude and consistency. Our study makes a compelling case for integrating neuroscience-inspired strategies into DNN design and learning mechanisms, encouraging further exploration of biologically plausible mechanisms to enhance the reliability and generalization of deep neural networks in real-world scenarios.

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

## A    Appendix

## B    Implementation Details

Here, we provide additional implementation details for CARoL.

### B.1    EMA Model Update

To accumulate knowledge across the different stages of training and avoid confirmation bias, CARoL maintain an additional exponentially moving averaged (EMA) model Tarvainen & Valpola (2017), $f_{\bar{\theta}}$, which averages the model's weights during training.

$$\bar{\theta} \leftarrow \alpha\bar{\theta} + (1 - \alpha)\,\theta, \quad if \ \ r_F > u \sim U(0, 1) \tag{7}$$

where $\alpha$ is the decay parameter which is set to 0.99. CARoL employs a stochastic update strategy for the EMA model, whereby instead of updating the weights of the EMA model, $\bar{\theta}$, at each iteration with the weights of the learning model, $\theta$, we sample a variable $u$ from a uniform distribution (between 0 and 1) and update the EMA model if $u$ is less than the update frequency parameter $r_F$. Additionally, to enforce the EMA model to aggregate more information from the early phase of training, where the model is less prone to memorization, we use a cosine ramp-down function for the update frequency:

$$r_F = r_{min} + (r_{max} - r_{min}) \cdot \cos(\frac{e \cdot \pi}{E}) \tag{8}$$

where $r_{max} = 1$, $r_{min} = 0.05$, $e$ is the current epoch and $E$ is the total number of epochs. This ensures that the update frequency of the EMA model is progressively reduced during training (Figure 6 left).

### B.2    Uniform Selection Across Classes

To encourage a more uniform selection of samples in the consistent set across classes, we take an average of the number of samples for each class in $\mathbb{C}$ and increase the number of samples for underrepresented classes in the consistent set. For this purpose, we assign a score to each unselected sample (with index i) of the under-represented class $= ((1 - \boldsymbol{c}_i) \cdot f(\boldsymbol{X}_i; \bar{\theta}))$ and select samples with low scores so that the number of samples for the class is equal to the average number of samples. This provides a principled way to use both prediction consistency and error magnitude to determine the quality of supervision.

### B.3    Error Memory

CARoL maintains an average error memory for each class, $\boldsymbol{e} \in \mathbb{R}^K$, by computing the exponential moving average of the EMA-model's average cross-entropy loss during the training trajectory:

$$\boldsymbol{e}_k \leftarrow \alpha\,\boldsymbol{e}_k + (1 - \alpha)\frac{1}{n_c}\sum_{i=1}^{N}(\mathbb{I}(y_i = k) \cdot \ell_{ce}(f(\boldsymbol{X}; \bar{\theta}), \boldsymbol{y}) \tag{9}$$

where $\alpha$ is the decay parameter set to 0.99.

To maintain the stability of error memory and prevent the statistic from being skewed toward large errors from noisy labels, we calculate the per-class average, $\mu_k$, and standard deviation, $\sigma_k$, of the loss values of the class samples in the batch and update $\boldsymbol{e}_k$ with errors values less than $(\mu_k + \rho \cdot \sigma_k)$ where $\rho$ is a hyperparameter.

### B.4    Dynamic Consistency Threshold

To account for the increase in consistency values as training progresses and the model makes more predictions, we also employ a dynamic consistency threshold $\tau$ using a sigmoid ramp-up function, similar to (Sarfraz et al., 2021; Laine & Aila, 2016) (Figure 6 right):

$$w(e) = \exp\left(-0.65(1 - \frac{e}{E})^2\right) \tag{10}$$

Hence, effectively at each epoch, the consistency threshold is given by $\tau \cdot w(e)$.

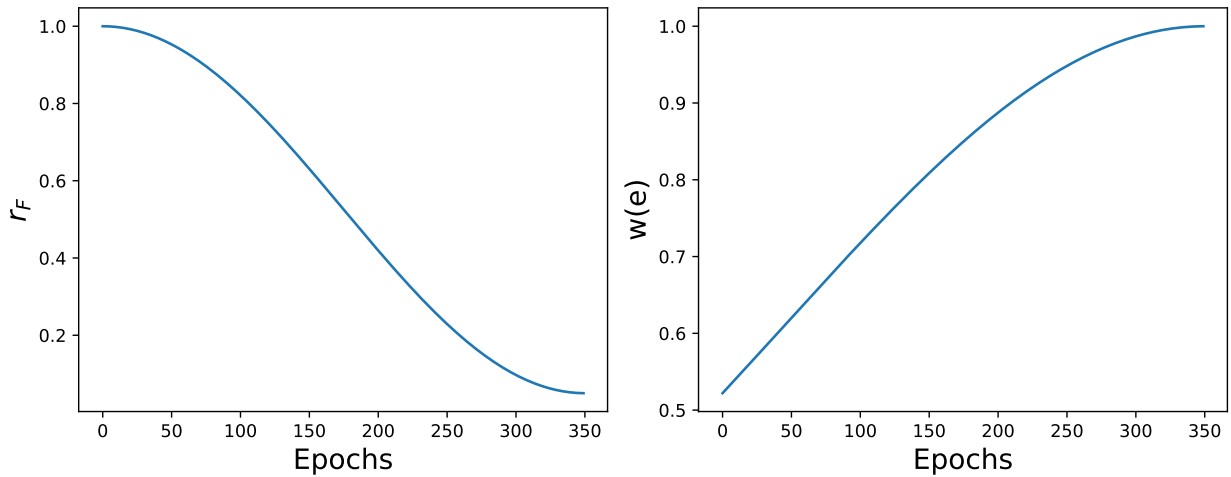

Figure 6: Ramp functions used for dynamic learning.

Table 5: Selected hyperparameters for the different settings.

| DATASET | NOISE (%) | $\tau$ | $\gamma$ | $\rho$ |
|---|---|---|---|---|
| | SYMM-20 | 0.3 | 3.0 | 2.5 |
| CIFAR-10 | SYMM-50 | 0.3 | 3.0 | 2.5 |
| | SYMM-80 | 0.3 | 0.75 | 0.1 |
| | SYMM-20 | 0.1 | 3.5 | 1.0 |
| CIFAR-100 | SYMM-50 | 0.1 | 1.5 | 0.3 |
| | SYMM-80 | 0.1 | 0.5 | 0.05 |
| | SYMM-20 | 0.3 | 1.0 | 1.0 |
| TINY-IMAGENET | SYMM-50 | 0.3 | 0.5 | 0.1 |
| | ASSYM-45 | 0.6 | 0.1 | 0.1 |
| WEB-AIRCRAFT | - | 0.1 | 1.0 | 2.5 |
| WEB-BIRD | - | 0.1 | 1.0 | 2.5 |
| WEB-CAR | - | 0.1 | 1.0 | 2.5 |

### B.5 Dynamic Weighting

We employ a similar strategy to gradually shift the focus of learning from cross-entropy loss to knowledge distillation loss by increasing the $\beta$ value using the Sigmoid function mentioned above. At each epoch, we use $\beta = \beta_{max} \cdot w(e)$ where $\beta_{max} = 0.95$.

### B.6 FixMatch

To further learn from samples that are considered inconsistent, we adapt the FixMatch loss (Sohn et al., 2020) such that the EMA-model's predictions on the weakly augmented image, $\boldsymbol{p} = \arg\max_k(f_{\bar{\theta}}(\boldsymbol{X}))$ with confidence score $\boldsymbol{s}$, act as the pseudo labels. We then enforce consistency in predictions, by training the model to predict the same label as the EMA model when fed a strongly augmented version of the same image $\boldsymbol{X}''$:

$$\mathcal{L}_{SSL} = \sum_i \mathbb{I}(i \notin \mathbb{C}, \boldsymbol{s}_i > \tau_s) \cdot \ell_{ce}(f_\theta(\boldsymbol{X}''), \boldsymbol{p}_i) \tag{11}$$

where $\tau_s$ is the confidence threshold set to 0.6. This enables us to utilize the inconsistent samples effectively for learning.

Table 6: Effect of different hyperparameters on the performance of the model trained on CIFAR10 with 50% Symmetric Noise.

| $\rho$ | $\tau$ | $\gamma$ | Acc (%) |
|---|---|---|---|
| 1.0 | 0.1 | 1.0 | 95.08 |
| | | 2.0 | 95.32 |
| | | 3.0 | 95.29 |
| | 0.3 | 1.0 | 94.67 |
| | | 2.0 | 95.51 |
| | | 3.0 | 95.51 |
| | 0.5 | 1.0 | 94.71 |
| | | 2.0 | 95.78 |
| | | 3.0 | 95.64 |
| 2.5 | 0.1 | 1.0 | 95.14 |
| | | 2.0 | 95.17 |
| | | 3.0 | 95.10 |
| | 0.3 | 1.0 | 95.55 |
| | | 2.0 | 95.56 |
| | | 3.0 | 95.88 |
| | 0.5 | 1.0 | 94.97 |
| | | 2.0 | 95.69 |
| | | 3.0 | 95.83 |

## C   Training Details

For a fair comparison, we use an experimental setup similar to previous methods (Li et al., 2020; Karim et al., 2022). Unless otherwise stated, we use PreActResNet-18 (He et al., 2016), SGD Optimizer with 0.9 momentum, 5e-4 weight decay, and an initial lr rate of 0.02 with cosine annealing lr scheduler, 64 batch size, and apply random crop and horizontal flip as weak augmentation. For CIFAR-10 and CIFAR-100 datasets, we train for 350 epochs and use CIFAR10 AutoAugment (Cubuk et al., 2018) policy as strong augmentation. For Tiny-ImageNet, we train for 150 epochs and use the ImageNet policy as a strong augmentation. We use a warm-up period of 30 epochs for all of our experiments and start tracking the predictions and loss values from Epoch 5. Following previous works (Li et al., 2020; Karim et al., 2022) we report the best accuracy for CIFAR10 and CIFAR100 datasets (the ablation study shows that there is a minor difference between the best performance and the last accuracy for CARoL; hence, it effectively mitigates forgetting) and for TinyImageNet, following (Han et al., 2018; Karim et al., 2022), we report the best and average accuracy for the last 10 epochs. For Web datasets, we follow the settings in (Sheng et al., 2024) and use a pre-trained ResNet-50 model with a learning rate of 0.005, batch size 32, and use SGD optimizer with weight decay 5e-4 and cosine annealing scheduler. We train the model for 250 epochs.

### C.1   Hyperparamter Tuning

A small clean validation set (5% of the training data) is used to finetune the hyperparameters with the best accuracy as the metric of selection. Tuning CARoL primarily involves tuning the threshold parameter $\tau \in \{0.1, 0.3, 0.6, 0.8\}$, the scaling parameter $\gamma \in \{0.5, 0.75, 1.0, 1.5, 2.0, 2.5, 3.0, 3.5\}$ and loss filtering $\rho \in \{0.05, 0.1, 0.3, 1.0, 2.5\}$. The parameters selected for each setting are provided in Table 5. For all experiments, we use $\tau_s$=0.6, except for CIFAR-100 80% where we set it to 0.2 to take into account the low confidence of the model when learning under such a high degree of noise and dataset complexity (with $\tau$=0.6, it achieves 93.61%).

### C.2   Sensitivity to Hyperparameters

Table 6 further shows that our method is not highly sensitive to a specific choice of hyperparameters and performs consistently across a wide range of parameters. Importantly, the parameters exhibit a synergistic

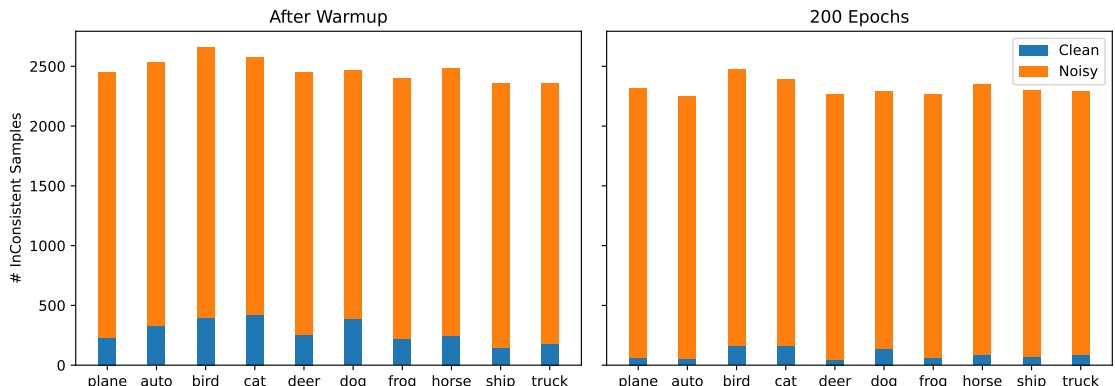

Figure 7: The distribution of clean and noisy labels among the samples from each class that are considered inconsistent. CARoL effectively distinguishes the vast majority of noisy samples as inconsistent.

relationship, enabling us to keep certain parameters constant while tuning others, which greatly simplifies the process of hyperparameter optimization.

## D   Baselines Considered

Depending on the availability of results in similar settings, we compare CARoL with the following state-of-the-art methods: M-Up (Zhang et al., 2018), PCIL (Yi & Wu, 2019), JPL (Kim et al., 2021), MOIT (Ortego et al., 2021), DMix (Li et al., 2020), ELR+ (Liu et al., 2020), UniCon (Karim et al., 2022), Decoupling (Malach & Shalev-Shwartz, 2017), MentorNet (Jiang et al., 2018), Co-teaching+ (Yu et al., 2019), M-correction (Arazo et al., 2019), NCT (Sarfraz et al., 2021).SOP (Liu et al., 2022), RML-Semi (Li et al., 2024) SED (Sheng et al., 2024).

## E   Distribution of Consistent and Inconsistent Samples

To further expand upon our analysis in Section 8, we examine how CARoL partitions training samples into consistent and inconsistent sets when training on CIFAR-10 with 50% label noise. Figures 5 and 7 illustrate the class-wise distribution of clean and noisy samples in both sets, measured after the warm-up stage and after 200 epochs.

As shown in Figure 5, the vast majority of clean samples are classified as consistent throughout training, with only a small fraction of noisy samples entering this set. In contrast, Figure 7 shows that most noisy samples are reliably identified as inconsistent even in the early phase, and this separation is further sharpened as training progresses. This demonstrates that CARoL not only minimizes false negatives (noisy samples mistakenly treated as consistent) but also preserves a high recall of clean samples.

Together, these results confirm the efficacy of our selection strategy: by leveraging prediction consistency and class-aware thresholds, CARoL achieves a robust separation of clean and noisy samples across classes. This principled partitioning prevents memorization of noisy labels and ensures that learning signals are drawn primarily from reliable samples, thereby mitigating forgetting throughout training.

## F   Comparison with SELFIE

While SELFIE (Song et al., 2019) also uses the consistency of previous predictions, there are several fundamental differences. It uses consistency to correct the labels of the samples while still using the small-loss trick to distinguish clean and noisy samples (assuming access to the noise rate, which is not a priori, and inferring it precisely using cross-validation is infeasible, as the range of possible values is huge).

In contrast, our study presents consistency in predictions as a more robust and reliable cue to identify clean and noisy samples, providing additional advantages, including estimating the transition matrix, correcting the labels, being more robust to different kinds of noise, etc. Therefore, our method uses consistent predictions to segregate clean and noisy samples instead of using the small-loss criterion.

Our main contributions are as follows: (1) A principled approach for mimicking error sensitivity modulation in the brain, whereby the model uses the history of predictions and loss values to modulate the per-sample sensitivity so that it learns more from small, consistent errors compared to abrupt, large errors. (2) Based on the characteristics of the sample, different learning mechanisms are utilized, i.e., higher weight for supervised learning of consistent samples and consistency regularization using SSL on inconsistent samples. (3) Dynamic shift in learning from supervised to SSL as training progresses; and (4) CA-MixUp loss that utilizes label refinement to correct the labels of inconsistent samples.

Our approach shows considerably higher performance: SELFIE achieves 86.5% accuracy on 40% symmetric noise (highest considered) on CIFAR10, which is considerably lower than CARoL even on double the noise (93.8% on 80% noise).

## G   Comparison with Robust Curriculum Learning (RoCL)

RoCL (Zhou et al., 2020) focuses on the unreliability of instantaneous loss, which exhibits high variance across epochs and leads to poor clean-label detection under severe noise. To reduce this variance, RoCL applies the small-loss criterion not on instantaneous loss but on EMA-smoothed losses, retaining small-loss filtering as its primary mechanism. Consistency in RoCL is defined as the discrepancy between the current model's predictions and those of an EMA model across multiple augmentations, and it is used only to identify samples with reliable pseudo-labels during the self-supervised phase. CARoL, by contrast, makes consistency the core learning signal. It maintains explicit memories of past predictions and classwise error statistics, defines consistency as the fraction of predictions matching the training label across epochs, and leverages this for principled error modulation, adaptive thresholds, and multiple learning pathways. This allows CARoL to utilize all training samples effectively, rather than relying primarily on small-loss filtering.

CARoL demonstrates significantly better performance compared to RoCL, showcasing the effectiveness of the selection criterion and learning mechanisms employed in CARoL. For instance on CIFAR10 and CIFAR100 with 80% label noise, RoCL achieves 85.76% and 53.89% respectively compared to CARoL's significantly higher performance 93.8% and 66.0%.

In summary, while RoCL and CARoL are motivated by related intuitions, our approach introduces a principled redefinition of consistency, a memory-based mechanism for error sensitivity modulation, and multiple learning pathways that allow all samples to contribute effectively. These innovations lead to both new insights into training dynamics under noisy labels and significant empirical improvements, particularly under severe noise and on challenging datasets.

