# OpenReview forum: "Consistency Aware Robust Learning under Noisy Labels"
_TMLR — Accepted by TMLR_

### Review · Reviewer_tuFR · 2025-08-09

**Summary Of Contributions:**

## Summary:

The authors introduce an algorithm to tackle label noise using a metric called the consistency score. They explain the different loss functions they combine and benchmark the algorithm’s performance on both synthetic (CIFAR-10 and CIFAR-100 with noise) and real-world datasets.

## Strength:
1. Benchmarking on real-world noisy datasets strengthened the claim for SOTA.
2. The quality of the visualizations is good.

## Weakness:
1. The paper is not easy to follow. In sections 2,3, and subsections 4.2.4 and 4.2.5, it is not clear which of the ideas are the authors' and which are from related work.
2. The component analysis table 4 is not sufficient to see which Loss contributes the most to performance.
3. The results on the CIFAR 10 dataset are not favourable, but the authors don't explain. Does this have anything to do with the symmetry of label noise?

**Additional Comments:**

n/A

**Audience:**

Yes

**Audience Explanation:**

The idea of using instance-level consistency during training to identify noisy labels is interesting.

**Claims And Evidence:**

No

**Claims Explanation:**

The authors have to make the motivations clearer and the novelty of the paper more convincing.

**Requested Changes:**

1. Fix the issues mentioned as weaknesses
2. Consider adding a Motivation section/subsection at the start, or a Discussion section at the end, to make the paper clearer

---

> ### Author Response · Authors · 2025-09-02
> **Author's response to Reviewer tuFR (1/2)**
>
> We sincerely thank the reviewer for the feedback and suggestions. We appreciate the recognition of our real-world benchmarking and visualizations, and we acknowledge the need to make our motivations and contributions clearer. In response, we have clarified the novelty of CARoL by explicitly highlighting its key contributions in the Introduction and adding a short Overview subsection before Section 4. We have also expanded the ablation study to include $\mathcal{L}_{CA}$ alone, which further validates the biological insights underlying our method. We also show the class-wise distribution of clean and noisy labels among the inconsistent samples (Figure 7, Section E) . This figure demonstrates that CARoL reliably identifies the vast majority of noisy samples as inconsistent, while minimizing false negatives, further validating the efficacy of our selection strategy. Below, we address each of the reviewer’s concerns in detail.
>
> > Biological Motivation and Key Contributions
>
> We thank the reviewer for the suggestion to make the motivations and contributions clearer. Our intention in Sections 2–4 was to present both the biological intuition that inspires CARoL and the related work that provides context. To improve clarity, we have now:
>
> - Added a Overview paragraph at the start of Section 4 (Methodology) that explicitly summarizes which components are novel.
> - Added a concise list of key contributions and insights in the Introduction to highlight our core findings.
>
> **Key Contributions of CARoL:**
>
> - **Consistency-guided learning:** CARoL maintains an explicit memory of model predictions throughout training and defines consistency as the percentage of predictions that match the training label. Unlike prior approaches that rely on instantaneous loss, CARoL uses this temporal consistency as the primary criterion to distinguish reliable vs. noisy samples.
>
> - **Class-adaptive thresholds:** We propose a principled mechanism that dynamically adapts consistency thresholds per class by comparing classwise averages with the global mean, thus accounting for class difficulty and class-specific noise prevalence.
>
> - **Error sensitivity modulation ($\mathcal{L}_{CA}$):** A biologically inspired loss that down-weights inconsistent samples based on error magnitude relative to classwise statistics, suppressing memorization while still leveraging difficult but informative samples.
>
> - **Multiple learning pathways:** Beyond standard supervised learning, CARoL integrates (i) FixMatch-style SSL to repurpose inconsistent samples using EMA-guided pseudo-labels, (ii) a consistency-aware MixUp that interpolates samples while mitigating noise, and (iii) a KL-divergence regularizer scheduled to favor early-phase predictions, which often align with true labels before memorization.
>
> **Key Findings**
>
> Our analysis of how the consistency metric evolves during training leads to several important insights:
> - **Clean samples remain consistent**: Clean samples maintain high prediction consistency aligned with their labels throughout training.
>
> - **Noisy samples show delayed consistency**: Noisy samples begin with low consistency, which gradually increases as the model memorizes the noisy label.
>
> - **Semantic similarity drives memorization speed:** Memorization occurs earlier when noisy labels are semantically similar to the true label (e.g., “dog” mislabeled as “cat”) compared to semantically distant classes (e.g., “dog” mislabeled as “car”).
>
> - **Early predictions are often correct:** Notably, early predictions for noisy samples frequently match the true label, making the early training phase a valuable surrogate for ground truth.
>
> These findings motivate CARoL’s design:
> - The EMA model is updated to retain a stronger bias towards early-phase predictions, preserving this reliable information.
> - Class-adaptive thresholds ensure noise patterns are handled differently per class.
> - All training samples are utilized—consistent samples directly for supervised learning, inconsistent samples via error-modulated weighting, SSL, and MixUp—rather than discarding noisy ones.
>
> In summary, CARoL redefines and exploits consistency in a fundamentally different way from prior work, yielding principled mechanisms for error sensitivity modulation and robust multi-pathway learning.

---

> ### Author Response · Authors · 2025-09-02
> **Author's response to Reviewer tuFR (2/2)**
>
> > On Component Analysis
>
> We have expanded the ablation to include $\mathcal{L}_{CA}$ in isolation (Table 4, row 3).  Results confirm its central role in CARoL’s motivation:
>
> - $\mathcal{L}_{CA}$ substantially improves stability over standard training (e.g., last-epoch accuracy +18.1% at 50% noise, +16.7% at 80%), highlighting its effectiveness in down-weighting inconsistent samples and suppressing memorization.
> - However, without SSL or CAM, inconsistent samples are not fully utilized for learning, limiting representation quality.
> - When combined with $\mathcal{L}_{KD}$, the benefits are amplified, as EMA-guided regularization complements consistency-based weighting.
>
> We would be happy to expand this further with additional combinations or dataset settings if this would help the reviewer assess the individual contributions more comprehensively.
>
> > On CIFAR-10 Results
>
> We respectfully note that CARoL performs strongly on CIFAR-10, achieving 93.8% accuracy even with 80% symmetric noise—very close to the clean-label ceiling. The relatively small margins between methods on CIFAR-10 are due to the dataset’s simplicity (low class overlap), where most SOTA methods already saturate performance. The advantages of CARoL are more pronounced on CIFAR-100, Tiny-ImageNet, and Web datasets, which involve higher semantic overlap and more realistic noise distributions, making them more challenging benchmarks for robust learning.
>
> We hope to have addressed the reviewer’s concerns, and we are happy to engage further or provide additional results to reinforce confidence in our contributions.

---

### Review · Reviewer_3oSY · 2025-08-12

**Summary Of Contributions:**

The paper studies the problem of noisy-label learning. Particularly, the paper proposes to use an instance-level consistency metric to identify noisy labels and then apply semi-supervised learning and mixup to learn the signals. The paper shows that the proposed method achieves improvements on datasets such as CIFAR10 and CIFAR100.

**Audience:**

Yes

**Audience Explanation:**

Noisy-label learning is a real-world problem that has a rich literature.

**Claims And Evidence:**

Yes

**Claims Explanation:**

The paper adopts a clear intuition and conducts empirical studies to verify the claim. However, such observations/intuitions have already been studied in [1]. Although some detailed designs are different, the paper's approach largely overlaps with [1] (e.g., using a metric to identify noisy labels based on time consistency, and also utilizing semi-supervised learning). Therefore, my biggest concern is how this paper is different from [1], how the findings for the noisy-label training dynamics differ from [1], and what improvements have been made both in terms of the algorithm design and empirical performance.

[1] Robust Curriculum Learning: from clean label detection to noisy label self-correction

The final loss contains four terms, and the paper uses beta to trade off between the first two terms following a specific schedule. Can the authors provide more justifications for the design of the schedule? Also, why don't we need additional trade-off factors for the other terms in the losses?

Minor:
Appendix: "Table 6: ... CIDAR10"

**Requested Changes:**

Please address the comparison with [1].

---

> ### Author Response · Authors · 2025-09-02
> **Author's Response to Reviewer 3oSY (1/2)**
>
> We sincerely thank the reviewer for their thoughtful and constructive feedback.
>
> > Comparison with Robust Curriculum Learning: from clean label detection to noisy label self-correction
>
> While both methods share the high-level intuition that accumulated signals are more reliable than instantaneous ones, our study introduces fundamental conceptual, algorithmic, and empirical differences. Below, we carefully outline how CARoL differs from RoCL:
>
> **Conceptual Focus:**
>
> -  RoCL: primarily relies on the small-loss criterion, applied on an EMA of per-sample loss; consistency is used only to obtain pseudo-labels for self-supervised learning. They proposes a curriculum learning approach which smoothly transitions between supervised learning on clean samples and self supervision using pseudo labels on noisy samples.
>
> - CARoL: makes consistency itself the central criterion. We define it as the fraction of predictions aligning with the training label across epochs, tracked via an explicit prediction memory. Importantly, CARoL draws inspiration from the dynamics of error driven learning in the brain and modulates the sensitivity based on the error magnitude magnitude and the consistency of predictions. It maintains an explicit memory of past predictions and per class error statistics, to inform learning and employs multiple learning pathways to learn from all the training samples. This enables principled error sensitivity modulation and classwise adaptation, rather than only serving consistency as a correction tool.
>
>
> **Algorithmic Distinctions:**
>
> - RoCL notes that instantaneous loss suffers from high variance across training epochs and is inaccurate for clean data detection under high noise ratio and randomness in DNN training. To address the high invariance issue, they show that the exponential moving average (ema) of the instantaneous loss shows lower variance and hence provides a more reliable signal. **They still apply the low loss criterion to detect clean samples**, but instead of applying it on the instantaneous loss, they apply it on the ema of the per sample loss.
>
> - Additionally, to address the challenges of obtaining a reliable pseudo label for training on noisy samples with self supervision loss, they posit that a model output tends to be an accurate pseudo label if it remains consistent over training samples and across different augmentations of the samples. Therefore they define consistency loss of a sample x at step t, as the discrepancy of the model output between step t and t-1 on x and its m different data augmentations.  To tackle the challenge of model’s output becoming too similar between step t and t-1, they use an ema of model parameters as a proxy for model’s prediction at an earlier time step. The pseudo labels are then obtained for samples with low discrepancy loss between the model's prediction on sample x and its m augmentations compared to the output of the ema model.
>
>
> - To summarize, **RoCL still uses the low loss criterion to identify clean samples**, and maily focuses on reducing the variance in loss signal by using an ema of sample losses instead of instantaneous losses. It **defines consistency as the discrepancy between the current model’s and ema model’s output on different augmentations of a sample x**, and only uses consistency **to obtain correct pseudo labels for noisy labels** during second phase of their curriculum learning.
>
> This is fundamentally different from how CARoL defines and utilizes consistency to learn under noisy labels. CARoL maintains an explicit memory of a model's predictions throughout the training trajectory and **defines consistency as the percentage of predictions which aligns with the training label**.
>
> We provide several **key insights** from how the aforementioned consistency metric evolves during training for noisy and clean labels:
> - clean samples maintain high prediction consistency and alignment with the training label throughout training, while noisy samples initially exhibit low consistency values, which gradually increase as the model begins memorizing the noisy label.
>
> - The onset of memorization depends on how semantically similar the noisy label is to the true label, i.e  memorization occurs earlier for noisy labels that are semantically similar to the true label, for instance memorization will happen earlier if a dog image is wrongly labeled as cat compared to if it is wrongly labeled as a car.
>
> - Notably, early predictions for noisy samples often align with their true labels.
>
> These observations suggest that if memorization can be avoided, the model’s predictions during the early phase of training can serve as a good surrogate for true labels, providing a strong incentive to bias learning towards this phase. And importantly, they suggest that we can use the consistency metric itself to estimate if a sample is noisy or clean. Importantly, we must account for the nature of noise in each class.

---

> ### Author Response · Authors · 2025-09-02
> **Author's Response to Reviewer 3oSY (2/2)**
>
> Based on the afforementioned insights, CARoL employ an EMA model, whose update strategy is defined such that it aggregates more information from the earlier phase of training, to maintain the prediction memory.
>
> - **Dynamic classwise thresholds:** Instead of using a single threshold value, we dynamically adapt the consistency threshold for each class based on the classwise average consistency compared to the global average consistency.
>
> - **Error-aware weighting:** Different from RoCL, all the training samples contribute to the supervised loss, with the important distinction that the contribution of losses belonging to inconsistent samples are modulated depending on their error magnitude compared to the mean class wise loss statistics.
>
> - **Multiple learning pathways:** To further maximize learning from all the samples, we adapt the FixMatch loss such that the predictions of EMA model on weakly augment image act as pseudo labels for training the model on strongly augmented image. Consistency aware MixUp loss is proposed to leverage all samples while mitigating the effect of noisy labels. Finally, to promote consistency and provide regularization from earlier phase of training, we use the KL divergence loss with a schedule such that the model relies more on imitation learning in the later phase.
>
> **Empirical Comparison:**
>
> CARoL demonstrates significantly better performance compared to RoCL, showcasing the effectiveness of the selection criterion and learning mechanisms employed in CARoL. For instance on CIFAR10 and CIFAR100 with 80% label noise, RoCL achieves 85.76% and 53.89% respectively compared to CARoL’s significantly higher performance 93.8% and 66.0%.
>
> In summary, while RoCL and CARoL are motivated by related intuitions, our approach introduces a principled redefinition of consistency, a memory-based mechanism for error sensitivity modulation, and multiple learning pathways that allow all samples to contribute effectively. These innovations lead to both new insights into training dynamics under noisy labels and significant empirical improvements, particularly under severe noise and on challenging datasets.
>
> We have also added RoCL to our related work in the revised manuscript and added a detailed discussion section in Appendex (section G)
>
> > Justification of the $\beta$ schedule
>
> $\beta$ ramp-up is standard in semi-supervised/consistency regularization. It prevents early dominance of knowledge distillation loss when EMA model is unreliable. Initially, higher weightage is given to the supervised loss as the model starts from random initializations and early training focuses on learning generalization patterns and is less vulnerable to memorization. As the model learns better representations, the training shifts more towards knowledge distillation as the EMA model can provide a stable signal and mitigate memorization in later phases of training where it is more susceptible to memorization. This aligns with our key insight that during the earlier phase of training, the model predictions of both clean and noisy samples tend to align with their true label and hence the model benefits from providing regularization from an EMA model whose update strategy is biased towards aggregating more knowledge from earlier phase.
>
> > No extra trade-off for SSL/MixUp
>
> Because the FixMatch loss is predominantly applied to inconsistent samples, we believe its  scale is naturally balanced. The consistency aware MixUp loss is designed such that it leverages all the samples for learning while mitigating forgetting by using the pseudo label from ema model for noisy samples and combining clean and noisy samples.These act as semi supervised losses in our formulation, enabling the network to learn from all the samples. The warm stage ensures that the model has learned sufficiently good representations and can leverage SSL losses efficiently. We do note, however, that it might be interesting to see the effect of a similar ramp up function on these losses.
>
> We hope to have addressed the main concerns of the reviewer and are happy to engage further to provide clarifications or supporting evidence to build the reviewer's confidence in the novelty and contributions of our work.

---

### Review · Reviewer_ZnWP · 2025-08-18

**Summary Of Contributions:**

The paper introduces Consistency-Aware Robust Learning (CARoL), a novel framework inspired by human brain error-based learning to enhance deep neural network robustness under noisy labels. By maintaining a memory of past predictions and errors, CARoL computes per-sample consistency scores and dynamically adjusts learning using a consistency-aware loss, semi-supervised learning, MixUp, and knowledge distillation. Empirical evaluations on CIFAR-10/100, Tiny-ImageNet, and real-world noisy datasets (Web-Aircraft, Web-Bird, Web-Car) demonstrate CARoL’s superior performance.

**Additional Comments:**

N/A

**Audience:**

Yes

**Audience Explanation:**

This paper proposes a method for addressing the noisy label learning problem and offers very interesting insights. Moreover, the paper is well-written and easy to follow.

**Broader Impact Concerns:**

There are no additional ethical concerns identified that require discussion.

**Claims And Evidence:**

No

**Claims Explanation:**

1. The authors state that “the brain prioritizes learning from small, consistent errors over large, abrupt ones,” motivating their approach to assign a high weight to samples in the consistent set in Eq. 4. However, the weighting scheme may lead to unintended outcomes. Specifically, for inconsistent samples ($i \notin C$ ), if the ratio $\frac{e_{y_i}}{\ell_{ce}}$  is sufficiently large, the weight for inconsistent samples could exceed 1, surpassing that of consistent samples. This contradicts the stated biological inspiration, as it may overemphasize noisy samples rather than prioritizing consistent ones. Additionally, in Eq. 4, why do the authors use the EMA model’s loss rather than the original model’s loss?

2. Equation 2 defines the class-adaptive threshold as $\tau_k = \frac{\frac{\sum_{i=1}^N \mathbb{I}(y_i = k) c_i}{n_k}}{\frac{\sum_{i=1}^N c_i}{N}} \cdot \tau$, where $\frac{\sum_{i=1}^N c_i}{N}$  is the global average consistency across all samples. Why is the global consistency average included in the denominator?

**Requested Changes:**

How is the consistency metric computed in Section 3?

The notation in Figure 4, such as $\mathcal{M}_p$  and $\mathcal{M}_e$ , is inconsistent with the main text's $M$  and $e$ . Additionally, blue and orange are used for clean and noisy samples without a clear justification. Adding an explicit explanation of the color scheme would improve clarity and reader understanding.

In Figure 5, the Warmup stage is referenced as the initial phase for evaluating the consistency-based selection criterion, but it is unclear whether this refers to a single epoch or multiple epochs, potentially confusing readers.  Additionally, Figure 5 would benefit from including the distribution of clean and noisy labels among samples identified as inconsistent for each class. This visualization could provide deeper insights into the selection criteria's behavior across classes.

Table 4 should include an experiment with only $L_{CA}$ to validate its core role in CARoL’s motivation for robust noisy label learning.

---

> ### Author Response · Authors · 2025-09-02
> **Author's response to Reviewer ZnWP**
>
> We sincerely thank the reviewer for the constructive and detailed feedback. We are happy to clarify the specific points raised and provide additional details, analyses, and visualizations to strengthen our submission.
>
> > On the weighting scheme in Equation 4
>
> We appreciate the reviewer’s observation. CARoL includes several guardrails that prevent inconsistent samples from outweighing consistent ones. Specifically:
>
> - Since CARoL effectively mitigates forgetting, the loss values of inconsistent samples remain sufficiently higher than class average losses, keeping the weighting ratio in check.
>
> - We use the EMA model’s losses because it provides more stable predictions, biased towards early-stage patterns where memorization is less likely. Both the prediction memory and error memory are updated from the EMA model to ensure smoother transitions.
>
> - Importantly, in implementation we clip \lambda_i to 1, so in rare cases where the weighting ratio becomes large, it never exceeds the weight of consistent samples. Normalization across the batch further ensures that consistent samples remain dominant in training. We apologize for the omission of this implementation detail in the paper, we have corrected it in the revised manuscript.
>
> > On Eq. 2 and the role of the global consistency average.
>
> The adaptive thresholding in Eq. 2 is designed to balance classwise differences in difficulty and noise prevalence. By scaling the classwise mean consistency against the global mean, CARoL adaptively sets stricter thresholds for classes with higher noise levels or error rates. This ensures that classes with higher inconsistency values are treated more cautiously with a propotionally higher consistency threshold.
>
> > On the computation of the consistency metric (Section 3).
>
> The consistency score is computed as defined in Eq. 1: the proportion of model predictions across training epochs that match the provided training label. This definition is consistent across Section 3 and Section 4.2.2 with the only difference that CARoL utilizes the EMA-model's predictions to populate the memory wheras for our base analysis in Section 3, we used a single model.
>
> > On Figure 4 notation and color scheme
>
> We thank the reviewer for pointing this out. In the revised manuscript, we have corrected the notation to match the main text and added an explicit explanation of the color scheme (blue = consistent/clean, orange = inconsistent/noisy) to improve clarity. Please note that these colors are arbitrarily chosen to visually contrast between noisy and clean samples and used consistently throughout the paper.
>
> > On the Warmup stage in Figure 5.
>
> As mentioned in Appendex section C (Training details), we use warmup period of 30 epochs for all of our experiments. We apologize that this was not clear in the main text. We have added this detail to the figure in the revised manuscript.
>
> > On including inconsistent-set distributions
>
> We thank the reviewer for the valuable suggestion and for helping us in strenghtening our study. We have added a complementary visualization (now Figure 7, Appendix) showing the class-wise distribution of clean and noisy labels among the inconsistent samples, both after warmup and at 200 epochs. This figure demonstrates that CARoL reliably identifies the vast majority of noisy samples as inconsistent, while minimizing false negatives, further validating the efficacy of our selection strategy. Please see section E in the revised manuscript.
>
> > On Table 4 and $\mathcal{L}_{CA}$ in isolation.
>
> Thank you for the suggestion, we agree that evaluating the effect of consistency aware loss  provides valuable insights. We have expanded our ablation study to include $\mathcal{L}_{CA}$ alone in Table 4. Results confirm its central role in CARoL’s motivation:
>
> - $\mathcal{L}_{CA}$ substantially improves stability over standard training (e.g., last-epoch accuracy +18.1% at 50% noise, +16.7% at 80%), highlighting its effectiveness in down-weighting inconsistent samples and suppressing memorization.
> - However, without SSL or CAM, inconsistent samples are not fully utilized for learning, limiting representation quality.
> - When combined with $\mathcal{L}_{KD}$, the benefits are amplified, as EMA-guided regularization complements consistency-based weighting.
>
> This addition directly validates CARoL’s design motivation and reinforces the importance of consistency-aware error adaptation in robust noisy-label learning.
>
> We hope our clarifications and additional results have addressed the reviewer’s concerns. We are grateful for the feedback and suggestions, which has helped us improve the clarity and rigor of our work. We would be happy to engage further and provide any additional details or analyses that could strengthen the reviewer’s confidence in our contributions.

---

### Author Response · Authors · 2025-09-02
**Summary of Changes in the Revised Manuscript**

We sincerely thank all reviewers for their valuable and constructive feedback. We greatly appreciate the time and effort invested in evaluating our submission. The feedback has helped us significantly improve the clarity, rigor, and presentation of our work. Below, we summarize the key changes made in the revised manuscript in response to the reviewers’ comments.

---

## **Summary of Revisions**

1. **Motivation and Key Contributions**
   - Added a dedicated Overview paragraph at the start of Section 4 to clearly distinguish CARoL’s novel components.
   - Expanded the Introduction to include a concise bullet list of **key contributions** and insights.

2. **Expanded the Ablation Study (Table 4)**
   - Expanded the ablation study to include $\mathcal{L}_{CA}$ in isolation
   - $\mathcal{L}_{CA}$ substantially improves stability over standard training (e.g., last-epoch accuracy +18.1% at 50% noise, +16.7% at 80%), highlighting its effectiveness in down-weighting inconsistent samples and suppressing memorization.
   - Updated Section 7 to discuss these findings explicitly.

3. **Distribution of Consistent and Inconsistent Samples (Section E)**
   - Added a new visualization (Appendix, Figure 7) showing the **distribution of clean and noisy samples in the inconsistent set**, complementing Figure 5 and providing deeper insights into sample selection behavior across classes.
   - Our analysis shows that most noisy samples are reliably identified as inconsistent even in the early phase, and this separation is further sharpened as training progresses.
   - This demonstrates that CARoL not only minimizes false negatives (noisy samples mistakenly treated as consistent) but also preserves a high recall of clean samples


We believe these revisions substantially strengthen the manuscript by clarifying CARoL’s novelty, highlighting its biological motivation, and expanding the empirical evidence supporting its design. We hope that the reviewers find our responses satisfactory and that the revised manuscript addresses their concerns. We thank the reviewers again for their valuable feedback and are happy to engage further or provide additional results if required.

---

### Decision · Action_Editor_daPQ · 2025-10-07

**Recommendation:** Accept as is

**Audience:**

Yes

**Audience Explanation:**

The work contributes to an active research area. The proposed CARoL framework provides practical value: it formalizes consistency memory as a robustness cue and integrates it into semi-supervised and MixUp-based learning. This approach should interest readers in robust machine learning.

**Claims And Evidence:**

Yes

**Claims Explanation:**

The authors present clear motivation, and extensive empirical validation. The revised manuscript addresses prior concerns by expanding ablation studies (Table 4) and providing detailed visualizations (Appendix E).

The additional discussion of guardrails in the weighting scheme (Equation 4) and adaptive thresholds (Equation 2) resolves earlier ambiguities. The experiments across CIFAR-10/100, Tiny-ImageNet, and Web datasets convincingly demonstrate robustness under severe label noise.

Overall, the claims are well supported by consistent empirical and analytical evidence.